# Dynamic Changes in the Distribution of Hydrocodone and Oxycodone in Florida from 2006 to 2021

**DOI:** 10.3390/pharmacy12040102

**Published:** 2024-06-28

**Authors:** Elena Lynn Stains, Akshay C. Patel, Jay P. Solgama, Joseph D. Hagedorn, Kenneth L. McCall, Brian J. Piper

**Affiliations:** 1Department of Medical Education, Geisinger Commonwealth School of Medicine, Scranton, PA 18510, USA; apatel09@som.geisinger.edu (A.C.P.); jsolgama@som.geisinger.edu (J.P.S.); bjpiper1@geisinger.edu (B.J.P.); 2Jefferson Einstein Hospital, Philadelphia, PA 19141, USA; 3Department of Pharmacy Practice, Binghamton University, Johnson City, NY 13790, USA; kmccall@binghamton.edu; 4Department of Pharmacy Practice, University of New England, Portland, ME 04103, USA; 5Center for Pharmacy Innovation and Outcomes, Forty Fort, PA 18510, USA

**Keywords:** opioids, PDMP, pill mill, ARCOS

## Abstract

Background: Florida, which led the country in terms of its number of opioid-prescribing physicians, was unique during the height of the opioid epidemic because of its lax prescribing laws and high number of unregulated pain clinics. Here, we address differences in the distribution rates of oxycodone and hydrocodone across Florida counties during the peak years of the opioid epidemic using an under-utilized database. Methods: The Washington Post and the United States Drug Enforcement Administration’s Automation of Reports and Consolidated Orders System (ARCOS) databases provided longitudinal oxycodone and hydrocodone distribution data in grams per county (2006–2014) and state (2006–2021). Grams of oxycodone and hydrocodone were converted into morphine milligram equivalents (MMEs). Results: There was a steep increase in oxycodone from 2006 to 2010, with a subsequent decline. In 2010, the average MME per person across Florida was 729.4, a 120.6% increase from 2006. The three counties with the highest MMEs per person in 2010 were Hillsborough (2271.3), Hernando (1915.3), and Broward (1726.9), and they were significantly (*p* < 0.05) elevated relative to the average county. Conclusions: The data demonstrated pronounced differences in opioid distribution, particularly oxycodone, between Florida counties during the height of the opioid epidemic. Legislative action taken between 2009 and 2011 aligns with the considerable decline in opioid distribution after 2010.

## 1. Introduction

Opioid distribution rapidly accelerated in the United States (US) in the 2000s. In 2013 and 2014, there were 10.7 million people who reported having abused pain relievers in the past year, with one-quarter (25%) of the pills being sourced directly from physicians [1]. Oxycodone and hydrocodone product misuse was particularly common among pain reliever abuse [2]. Deaths due to opioid overdose increased from over 8000 in the year 2000 to over 46,000 in 2018 [3]. The burgeoning trend of medical providers over-prescribing opioids to patients during this period is often cited as a driver of the opioid crisis in the US. Opioids are commonly prescribed to control moderate to severe acute pain [4]. Physicians usually prescribe these narcotics after major surgery or severe injury or even for chronic pain [4,5]. Opioid analgesics exert their effect by binding to mu (μ) opioid receptors in the central nervous system [6]. These drugs stimulate the inhibitory descending pathways involved in pain perception [6]. Some patients develop a tolerance for the medication, requiring a higher dose to achieve the same level of pain relief. The increased need for opioid medication can put patients at risk of overdose. Ingesting large amounts of opioids can lead to severe respiratory depression, leading to death [7]. Furthermore, overdose complications can include pulmonary edema, hypoxia, and aspiration pneumonia, directly increasing the risk of mortality [8,9]. Patients also face the risk of forming physical dependence, which may be accompanied by withdrawal symptoms if they lose access to the medication [9].

One of the key states leading the opioid epidemic in the US was Florida. The preponderance, 98, of the United States’ top 100 opioid-prescribing physicians in 2010 operated out of Florida [10]. During this year, the state reached a historic peak in opioid distribution and ranked 13 out of 50 states in terms of the opioid overdose death rate [11,12]. To deal with the erupting opioid crisis, the Florida government began to address the high distribution of opioids in the state by creating the prescription drug monitoring program (PDMP) in 2009, which required specific reporting from both the prescriber and pharmacist for each controlled substance prescription [13]. The PDMP requires information such as provider name and Drug Enforcement Administration (DEA) number, date filled and method of payment by the patient, and pharmacy name, address, and DEA number [13]. Two years later, Florida signed into law HB7095, which aimed to control rogue pain management clinics in the state by limiting providers’ ability to dispense opioids directly [14]. The law gave providers 10 days from its enactment to dispose of any controlled substances still in possession [14]. Law enforcement identified major contributors to direct controlled substance distribution and were empowered by this law to enter these businesses and ensure disposal [14]. This law also required that pharmacies be re-licensed with new requirements to prevent the dispensing of controlled substances through fraudulent methods, including invalid practitioner–patient relationships [14].

Here, we explore the opioid distribution across Florida counties during the peak distribution years of the opioid epidemic. This is the first examination of this state using data from the United States Drug Enforcement Administration’s Automated Reports and Consolidated Orders System (ARCOS), which was obtained by the Washington Post [15,16]. Florida’s opioid spike has been a popular topic in the news, and although some research has been undertaken on the effect of the state’s PDMP and pill mill legislation, more questions are still able to be answered with ARCOS [17,18]. The Centers for Disease Control (CDC) published data from IQVIA, which report the number of opioid prescriptions per capita at the county level from 2006 to 2020; however, these data are not divided by opioid type, do not include the dose or total amount of opioids given per prescription, and only comprise retail (non-hospital) pharmacy data [19]. It is well known that Florida providers prescribed a high amount of opioids at the height of the opioid epidemic—the state’s opioid dispensing rate was 87.6 prescriptions per 100 people in 2010 [19]. However, research has yet to discover more details about the nature of the prescription opioid boom in Florida—where distribution clustered (if at all), how high distribution rose at its peak, and how long it took for distribution to decrease after legislation was passed. We found specifics like these through ARCOS.

While opioid overdose deaths remain high in the US at present, with prescription opioids responsible for 24% of opioid deaths in 2020, providers and researchers continue to look for safer ways for opioids to be prescribed [20,21,22]. Past over-prescribing and the high distribution of opioids are popularly described as leading to deadly consequences [23]. This paper explores key details of the opioid distribution at the peak of the opioid crisis in Florida using an under-utilized database. ARCOS reports on the distribution of opioids, including the type of opioid and amount in grams, to not only large retail pharmacies but also independent pharmacies, hospitals, VA facilities, and Indian Health Services facilities. ARCOS also reports on the distribution to individual providers, a particularly intriguing factor in the state of Florida, where providers were once allowed to dispense opioids directly from their offices (i.e., without pharmacists) [24]. The standard ARCOS database, which reports on Schedule II and III substances with limited geographic resolution, has been used in some prior reports [11,25,26,27,28,29]. This report, which also includes the Washington Post ARCOS, adds to a more limited database of peer-reviewed investigations [30,31]. We analyzed this information for each county in Florida to compare easy-to-conceptualize areas of the state that may differ in population, urbanicity, demographics, and density of providers.

## 2. Materials and Methods

Data source: Data obtained from ARCOS and the Washington Post provided information on the longitudinal oxycodone and hydrocodone distribution [16,24]. We include data from the standard ARCOS dataset for 2006 to 2021 to provide further insight into the hydrocodone and oxycodone distribution beyond 2014. Data included from the Washinton Post ARCOS comprised the total amount of oxycodone and hydrocodone reported in Florida between 2006 and 2014. Data included from the standard ARCOS comprised the total amount of oxycodone and hydrocodone reported in Florida between 2006 and 2021. Data that were excluded contained repeat values that ARCOS erroneously reported. The standard ARCOS data report opioids distributed to several business types (pharmacies, hospitals, providers, etc.), while the Washington Post data include only opioids distributed to pharmacies. Unlike the standard ARCOS, the Washington Post data can be examined by county, which was of interest to this study.

Measures: Grams of oxycodone and hydrocodone were each converted into morphine milligram equivalents (MMEs) and totaled. Oxycodone was converted into MMEs by multiplying the dose by 1.5. Hydrocodone was converted into MMEs by multiplying the dose by 1 [32]. MMEs can be reported in any weight, including milligrams, grams, or metrics tons, depending on the weight of the dose being converted. For context, the daily dose of oxycodone or hydrocodone for moderate pain after laparoscopic surgery is 15 to 20 milligrams [33]. Twenty milligrams of oxycodone is equal to 30 MMEs. ARCOS has been validated by comparing it to a statePDMP, and the result was r = 0.985 [25]. The Washington Post ARCOS database has been used in previous research [34,35,36]. Each county’s (Appendix A) total MME was divided by county population during the given year according to the American Community Survey to obtain the MMEs per person [37]. The MME per person across counties and study years was plotted to assess the peak distribution year. We then designated three counties as the “top three counties” and three counties as the “bottom three counties” based upon their MMEs per person during the peak year, 2010. We created heat maps to visually represent each county’s MMEs per person and MMEs per pharmacy using RStudio. The procedures were approved by the IRBs of the University of New England and Geisinger as exempt.

Statistical analysis: A 95% confidence interval (1.96×SD) was determined, and counties were considered statistically different if they fell outside of this range. This was used to evaluate each county’s average MMEs per person across the study years, as well as the MMEs per person in the peak year.

## 3. Results

### 3.1. ARCOS 2006–2021

The ARCOS data showed a sharp increase (+215.3%) in oxycodone distribution from 2006 to 2010. There were 18.72 metric tons of oxycodone distributed in the state in 2010. Oxycodone distribution decreased by 69.1% from 2010 to 2014 and remained below peak levels, despite a small rebound from 2016 to 2017. By 2021, oxycodone distribution in the state was 1 MME (metric ton) lower than in 2006. Hydrocodone decreased by 58.5% from 2006 to 2021 (Figure 1a).

Throughout the study timeline, pharmacies remained the primary point of distribution for both oxycodone and hydrocodone (Figure 1b,c). Of the total weight of oxycodone distributed in 2006, 93.12% was distributed by pharmacies, 4.05% by hospitals, and 2.83% by medical practitioners. The percentage distributed by practitioners increased to 10.9% in 2009 and then dropped to nearly zero by 2011 (Figure 1b). In 2011, that distribution was then relocated to pharmacies (97.65%) and hospitals (2.11%). As for hydrocodone, 95.86% was distributed by pharmacies, 2.30% by hospitals, and 1.84% by practitioners in 2006. The percentage distributed by providers also decreased after 2009, remaining under 0.4% through 2021 (Figure 1c).

### 3.2. Washington Post ARCOS 2006–2014

The state distributed a total of 103,315.43 MMEs in kilograms of hydrocodone and oxycodone from 2006 to 2014. Florida saw a steady increase in the MMEs of oxycodone (+230.2%) from 2006 until it peaked in 2010. However, the MMEs of hydrocodone decreased (−11.5%) from 2006 to 2010. A total of 19,958.85 MMEs were distributed in 2010. Distribution subsequently declined for oxycodone (−70.2%) and hydrocodone (−7.2%) from 2010 to 2014.

When all the counties were averaged, there was more than a three-fold rise in oxycodone from 2006 to 2010 (Figure 2a). While the total amount of oxycodone in Florida increased from 2006 to 2010, there was great variability among counties. Dixie County distributed only 2.95 MMEs in 2006 and decreased this distribution by −10.2% to 2.65 MMEs by 2010. However, most counties increased their distribution substantially. Broward County’s distribution rose 147.5% from 1222.78 MMEs in 2006 to 3026.98 MMEs in 2010.

The MMEs per pharmacy followed a similar, yet less pronounced, pattern. From 2006 to 2010, there was a 152.7% increase in the MMEs of oxycodone and a 53.8% decrease after 2010. Hydrocodone declined slightly (−2.1%) from 2006 to 2014 (Figure 2b).

Florida’s mean MME per person across the study years was 490.0. Two counties’ mean MMEs per person were significantly (*p* < 0.05) higher than the state’s (490.0): Hernando (1019.2) and Hillsborough (1110.0). Nine were elevated above the mean: Baker (750.1), Broward (891.9), Charlotte (863.6), Okeechobee (844.2), Palm Beach (786.4), Pasco (940.5), Pinellas (915.8), Putnam (812.3), and Sarasota (769.5).

In 2010, the average MME per person in milligrams across the state was 729.4. This was a 120.6% increase from 2006, when the average MME per person in Florida was 330.7. The three individual counties with the highest MMEs per person in the peak year were Hillsborough (2271.3), Hernando (1915.3), and Broward (1726.9), which were significantly (*p* < 0.05) elevated relative to the state’s average in 2010. The three counties with the lowest MMEs per person in the peak year were Dixie (162.0), Lafayette (154.8), and Liberty (136.8). However, none of the three were significantly different from the state average in 2010 (Figure 3a).

The MMEs per pharmacy again followed a similar pattern. When comparing our top three and bottom three counties, most displayed the same pattern, with the exception of Dixie (Figure 3b). The MMEs per person correlated highly (r = 0.91) with the MMEs per pharmacy (Figure 4). The counties with the highest and lowest MMEs per person (Figure 5a) and MMEs per pharmacy (Figure 5b) at the peak of the distribution years are generally clustered together in the state.

There was a high correlation between the traditional ARCOS and the Washington-Post ARCOS (Appendix A).

## 4. Discussion

Obvious in these striking results are several key findings regarding the oxycodone and hydrocodone distribution in Florida. There was a steep increase in the distribution of these Schedule II opioids in the Sunshine State from 2006 to 2010, followed by a similarly large drop-off. The raw MME distribution more than doubled in the years leading up to the peak. The pronounced increase in the total MMEs was attributable to a rise in oxycodone, as hydrocodone distribution declined over this period. A drop in practitioner-related business activity was met by an increase in pharmacy-related business activity. When practitioner business activity fell after 2010 to account for around 1% of the total, pharmacy business activity saw a rise to about 98%. Because providers were no longer legally allowed to dispense opioids directly, pharmacies began to dominate almost all of the opioid business activity. The average MMEs per person across the state rose according to a similar pattern to the raw MMEs. However, when broken down by county, we saw that while some counties followed this uptick, others remained steady, with very limited distribution. A few counties, namely Broward, Hillsborough, and Hernando, saw a peak in MMEs per person far above the state’s average. The state’s mean MME per person in 2010 was 729—for context, 729 MMEs equal 729 milligrams of hydrocodone or 486 milligrams of oxycodone. If one oxycodone pill is 10 mg, that is more than 48 pills per Floridian in one year. This pattern seems to have some regional differences, as Hillsborough, Hernando, and Pasco are adjacent to each other on Florida’s west coast in the vicinity of Tampa (Figure 5a). Broward County is located on the opposite coast but contains major cities like Fort Lauderdale, Davie, and Hollywood (Figure 5a).

The top three counties, Hillsborough, Hernando, and Broward, are home to the cities of Tampa, Spring Hill, and Fort Lauderdale, respectively. The bottom three counties, Dixie, Lafayette, and Liberty, are located in the north of Florida and contain Cross City, Mayo, and Bristol, respectively. Hillsborough and Broward Counties each have populations of over one million, while Dixie, Lafayette, and Liberty Counties have some of the lowest populations in the state, at under 18,000 people. The top three counties have generally higher median incomes ($60,566, $50,280, and $60,922) than the bottom three counties ($41,674, $51,734, $39,121). The top counties also have lower poverty rates (11.9%, 12.5%, 11.0%) than the bottom counties (23.2%, 20.7%, 21.2%) [37]. The majority of the distribution existing in higher-income counties begs the question of why this may be. Counties with higher average incomes may have easier access to medical care for chronic conditions. People living in these counties may have the means and time to visit a doctor, increasing their likelihood of being prescribed an opioid for chronic pain.

Another reason for the top three counties distributing such disproportionate numbers of opioids is likely because they were often the sites of “pill mill” clinics. One infamous pain clinic arrest was that of Sylvia Hofstetter, who operated pill mills in Broward County and Tennessee, helping to distribute more than 11 million pills of oxycodone, oxymorphone, and morphine [38]. The Florida Department of Law Enforcement reported in 2010 that Tampa alone contained 103 pain clinics [39]. Similarly, Broward County had over 150 pain clinics in 2009 [39]. Even some pharmacies were responsible for the high distribution of opioids—a total of 53 arrests were made at Glory Pharmacy in Hernando County, when police found they were not only accepting fraudulent scripts for controlled substances but also that the owners were trafficking oxycodone pills by the hundreds [40,41].

The second noteworthy finding is the great decline in MMEs and MMEs per person in the state after the peak in 2010. The health policies of the state at the time may offer some insights into why these numbers dropped. The sharp increase in opioid distribution in 2010 is thought to be a consequence of the many infamous “pill mill” physicians in Florida at that time [10]. In 2009, as opioid use rose, the state passed legislation creating a PDMP, known as E-FORCSE (Electronic-Florida Online Reporting of Controlled Substance Evaluation Program), to add safeguards against physicians maliciously prescribing controlled substances [13]. The program, which was not implemented until 2011, required dispensers of controlled substances to file a report each time a Schedule II, III, IV, or V substance was given to a patient [13]. To deal specifically with the vast number of pill mill physicians in Florida, the US DEA launched a multi-agency operation called Operation Pill Nation to seize clinics, arrest physicians and clinic staff, and, in some instances, seize assets such as exotic cars and weapons [42]. Immediately after the raids is when Florida legislators passed HB7095, barring physicians from dispensing Schedule II or III drugs directly out of their offices. A study of 43 Florida physicians who faced criminal action for their involvement in pill mills found that 25 (58%) of those physicians permanently lost their licenses to practice in the state [43]. However, 17 physicians kept their license for one year or more after being indicted [43].

The impact of Florida’s PDMP has been significant in reducing the over-prescription and misuse of prescription opioids in the state. Florida’s PDMP was associated with a 25% reduction in opioid overdose deaths in the state between 2010 and 2012 [44]. This study also found that the PDMP was effective in reducing the number of opioid prescriptions, the number of patients who received high doses of opioids, and the number of patients who received opioids from multiple providers [44]. Interestingly, Florida’s decline in opioid distribution after 2010 was not unique. Eufemio et al. found the same timing of decline in Delaware, Maryland, and Virginia [30]. Although the authors believed this decline in oxycodone and hydrocodone to be due to stricter regulation as well, these three states implemented PDMPs at scattered points in time (Delaware in 2012, Maryland in 2018, Virginia in 2003) without much temporal correlation with the decline in opioids [45,46,47]. This suggests the cause of the 2011–2014 decline in opioid distribution may be different among the four states.

However, there may have been an unintended consequence of the legislation—fentanyl has become a major problem in Florida, which is concerning, as it is more potent than other opioids [48]. According to the Florida medical examiners commission, in 2021, fentanyl-related deaths were 50 times higher than in 2010, the peak year of the opioid crisis [49]. Tampa and Ft. Lauderdale, cities in two of our top three counties, had some of the highest rates of fentanyl deaths in the state in 2021 [49]. A 2022 study interviewed 148 opioid users from three states to explore their switch from oral opioid use to injection of heroin and fentanyl [50]. Nearly half (48%) of the participants began with prescribed oral opioids [50]. The common thread in the participants’ accounts described by the authors was that when states began regulating prescription opioids more strictly, doctors stopped prescribing, and oral opioids became more scarce and therefore more expensive [50]. This lead to participants finding other, cheaper, and more available methods of securing opioids; therefore, they began using IV heroin and fentanyl [50]. Congruent with our data for Florida, these participants reported the scarcity of prescription opioids beginning in 2010 [50].

A key question is why there was this increase in oxycodone and not hydrocodone. The 2015 National Survey on Drug Use and Health found that hydrocodone products were the most commonly used and misused pain relievers in the US, contrary to what Florida’s data indicate [2]. However, a factor that may have affected providers in Florida is the FDA’s 2013 recommendation to change hydrocodone products from Schedule III controlled substances to Schedule II [51]. The anticipation of hydrocodone rescheduling may have contributed to providers choosing other less scrutinized pain relievers, like the previously uncontrolled, unscheduled tramadol [52]. This explanation is limited, however, as analyses in other states (Delaware, Maryland, and Viriginia) found that hydrocodone did increase from 2006 to 2010, although not as dramatically as oxycodone [30].

### Limitations

This study focused on characterizing the opioid distribution pattern for Florida over a fifteen-year period (2006–2021) using two similar datasets, each with their own strengths. The traditional ARCOS is updated approximately every six months but does not provide for a geographic resolution beyond that of the three-digit zip code. The Washington Post ARCOS does not provide any data beyond 2014 but can be analyzed to the level of the individual county or pharmacy. The correlation between the two datasets was extremely strong but not perfect (r^2^ = 0.98, Appendix A). The Washington Post ARCOS data were processed such that it removed repeated and null datasets before publishing them [53]. As a part of the cleaning of the data, the Washington Post removed shipments of opioids that did not make it to the consumer [53]. The association of less than 1.0 is likely due to these refinements in the Washington Post ARCOS.

The ARCOS database is more difficult to work with in terms of computation. The information is available in PDF format, which can make analysis more challenging than Excel format. The benefits of using ARCOS, however, is that it is very comprehensive compared to other sources. The ARCOS data also include Veterans Affairs data, as well as Indian Health Services data. The IQVIA data source omits this information. Also, the ARCOS data also include data from veterinarians [54,55]. Veterinarian-prescribed opioids could be provided for pets for pain management after surgical procedures, but owners could also use these medications. Although it cannot be proven that this occurred, it is a possible risk.

This study examined only the patterns for oxycodone and hydrocodone, and although these two are generally the most prescribed opioids, prior investigations have evaluated other opioids [11,25,56]. Further investigation may reveal the possible socio-economic, geographic, and political impacts that caused certain counties to be disproportionately affected by high distribution compared to others. Inherently, the ARCOS database may contain biases regarding the representation of race. Patients of color tend to experience different pain control regimens that are less aggressive compared to their Caucasian counterparts [57]. The data in ARCOS do not provide any information about patient demographics, making it difficult to assess any disparities.

Further research could be carried out to analyze some interesting trends noted in the standard ARCOS data. Figure 1a indicates a small increase in oxycodone distribution from 2016 to 2017. A clear reason for this was not easily uncovered. A further dive into the social effects of rapidly decreasing opioid distribution may explain the small rise; however, it is not obvious.

## 5. Conclusions

In conclusion, this study evaluated the pronounced temporal and geographic variation in oxycodone and hydrocodone distribution in a state that saw noteworthy levels of opioid use. We demonstrated the peak of distribution in 2010 and the subsequent fall. The timing of this fall can be temporally associated with Florida’s implementation of a PDMP and legal pursuit of pill mill medical providers. We also demonstrated the stark differences in the distribution between counties, with some (Broward, Hillsborough, Hernando) distributing far above the state average. These findings add to our understanding of the opioid crisis and may provide a valuable lesson for preventing future iatrogenic opioid epidemics in the US and globally.

## Figures and Tables

**Figure 1 pharmacy-12-00102-f001:**
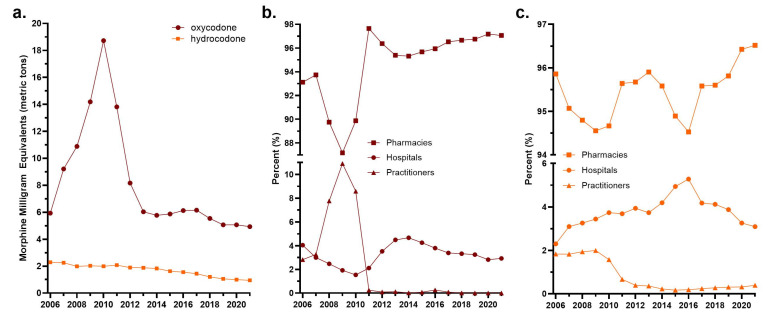
Drug Enforcement Administration’s Automated Reports and Consolidated Orders System (ARCOS). (**a**) Weight of oxycodone and hydrocodone in morphine milligram equivalents (MMEs) distributed in Florida from 2006 to 2021. Percent of (**b**) oxycodone and (**c**) hydrocodone distribution by business activity.

**Figure 2 pharmacy-12-00102-f002:**
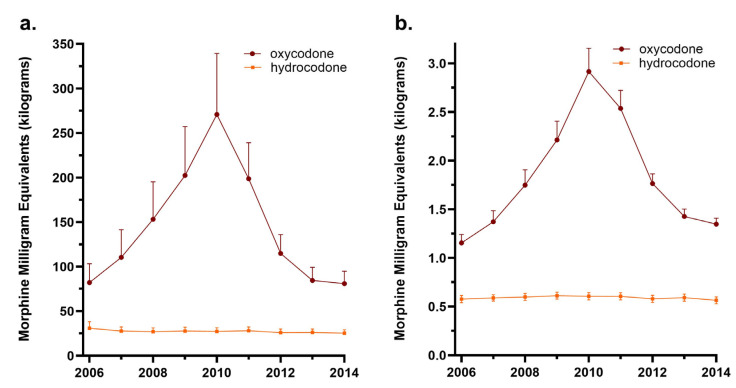
Total MMEs (**a**) and MMEs per pharmacy (**b**) of average Florida county for oxycodone and hydrocodone from 2006 to 2014 as reported to the Washington Post ARCOS.

**Figure 3 pharmacy-12-00102-f003:**
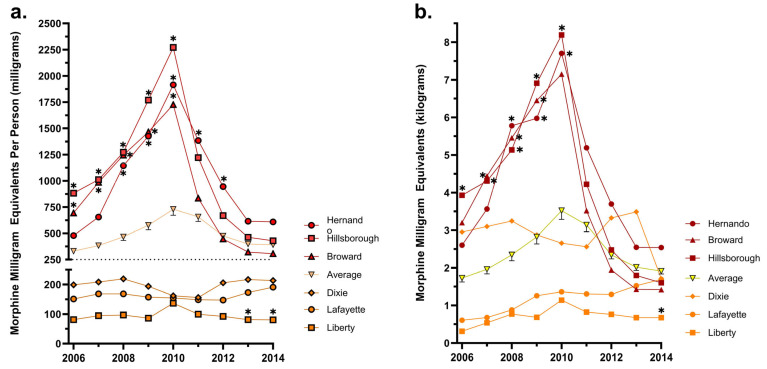
MMEs per person (**a**) and per pharmacy (**b**) in the top three (Hernando, Hillsborough, Broward) and bottom three (Dixie, Lafayette, Liberty) counties relative to Florida’s average as reported to the Washington Post ARCOS. Counties that were statistically different (*p* < 0.05) from the mean are marked with an asterisk.

**Figure 4 pharmacy-12-00102-f004:**
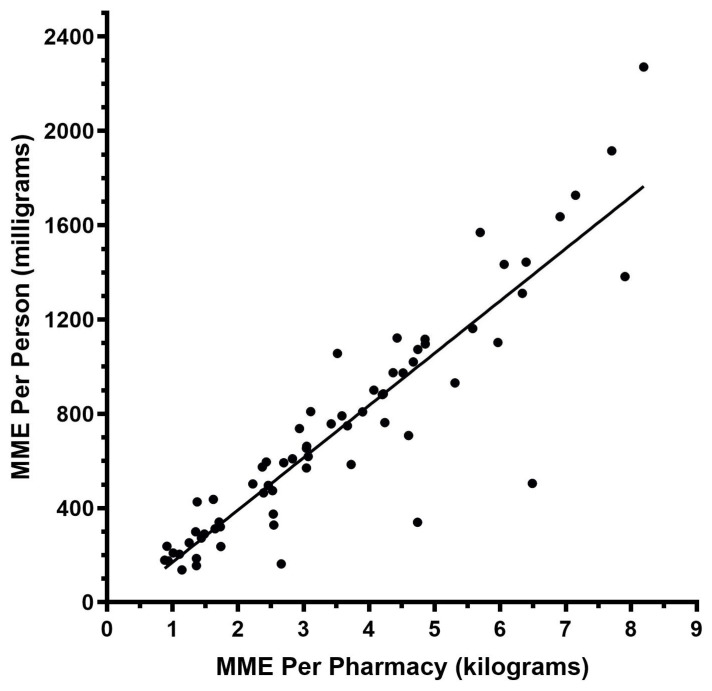
Correlation (r = 0.91) and linear regression of county-level data between MMEs per person and MMEs per pharmacy in the peak year, 2010, as reported to the Washington Post/Drug Enforcement Administration’s Automated Reporting and Consolidated Orders System (ARCOS).

**Figure 5 pharmacy-12-00102-f005:**
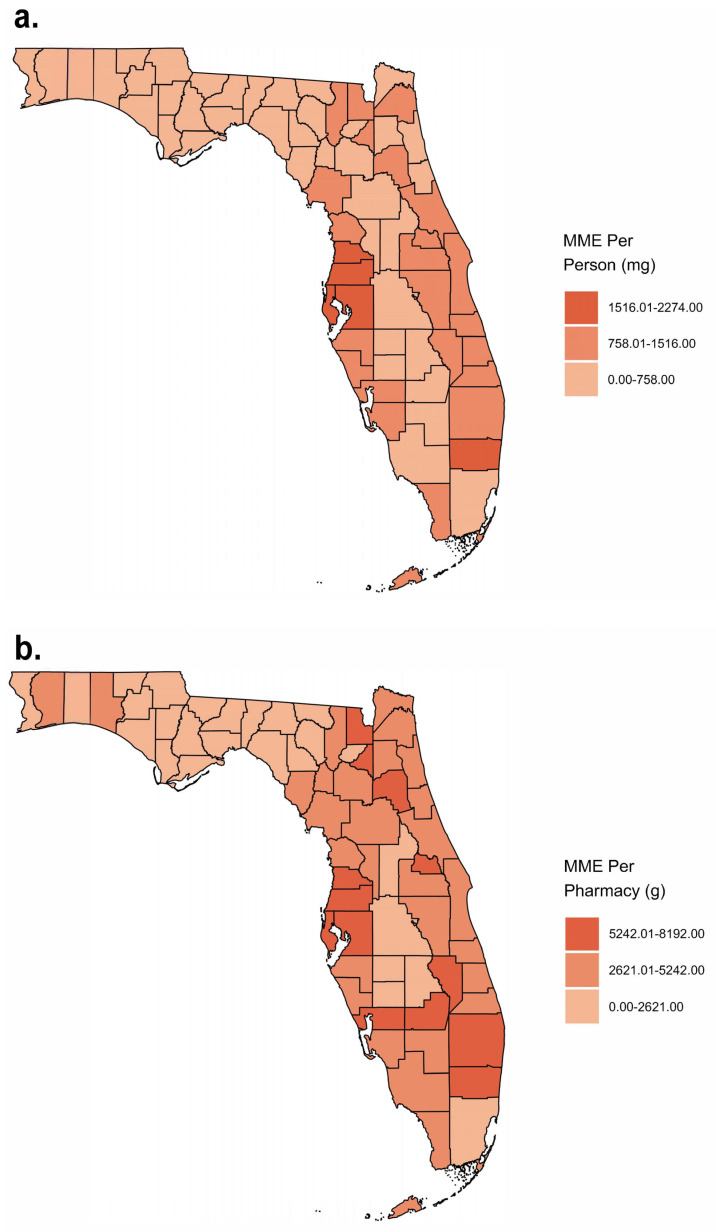
Heat maps representing the MMEs (milligrams) per person (**a**) and MMEs (grams) per pharmacy (**b**) per county in 2010 as reported to the Washington Post ARCOS.

## Data Availability

The ARCOS data are publicly accessible at https://www.deadiversion.usdoj.gov/arcos/retail_drug_summary/arcos-drug-summary-reports.html (accessed on 30 January 2022); the WaPo ARCOS data are publicly accessible at https://www.washingtonpost.com/graphics/2019/investigations/dea-pain-pill-database/ (accessed on 30 January 2022).

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
