# Peer review of "Dynamic Changes in the Distribution of Hydrocodone and Oxycodone in Florida from 2006 to 2021"

_pharmacy, 2024, doi:10.3390/pharmacy12040102_

Round 1

Reviewer 1 Report

Comments and Suggestions for Authors

The current study reports the rise in the use of oxycodone between 2006-2021 and its rise in year 2010 in the state of Florida. Overall, the report is well-written and contains useful information. However, some issues noted that need the attention of the authors.

Major:

1. In the Introduction section, some basic information about the pharmacology of oxycodone and hydrocodone needs to be provided. Notably, how these drugs can cause death due to overdose and what may be the cause of overdose and overconsumption. Drug availability may be one of the reasons.

2. Exclusion and inclusion criteria need to be provided in the Method section.

3. Section 3, the authors stated a 58% decrease in hydrocodone use. There is also some reduction for oxycodone as well and that needs to be stated.

4. Why are the authors focusing only on the practioners, and not considering pharmacies and hospitals? Changes in the distribution of each drug should also be included for the pharmacies and hospitals. It shows hospital's and pharmacies' distribution increased in recent years compared to 2006.

Minor:

1. It would be better to use the same color for oxycodone and hydrone and the same legends for all figures.

2. There should be a space between the word before a citation and the reference in the parenthesis.

Comments on the Quality of English Language

Some minor issues were noted and stated above.

Reviewer 2 Report

Comments and Suggestions for Authors

In the present manuscript, the authors addressed differences in distribution rates of oxycodone and hydrocodone across Florida counties during the peak years of the opioid epidemic using a novel database. In my opinion, the manuscript is interesting as understanding the temporal and geographic variation in opioid distribution may help expand our understanding of the opioid crisis and provide a valuable lesson for preventing future iatrogenic opioid epidemics.

Major comments:

Discussion:

The authors could discuss their results with those obtained by Eufemio et al. 2023.

Conclusions:

The conclusion section must not contain any references. Please move these lines to Discussion. The conclusion must approach the main findings; therefore, it needs to be rewritten.

Minor comments:

Introduction:

The authors could expand the introduction by adding more information about the Prescription Drug Monitoring Program and the Pill Mill legislation.

Materials and Methods:

The authors mention the use of a novel database, but at the same time indicate that it has already been used in previous research. Please be consistent and/or clearly highlight the novel nature of the database.

“There was a high correlation between the traditional ARCOS and the Washington-Post ARCOS (Supplemental Figure S1)”. In my opinion, this sentence should be carried over to the results.

Results:

“Figure 4. Correlation (r = .0.91)” change to “Figure 4. Correlation (r = 0.91)”.

Additional comments:

Acronyms/Abbreviations/Initialisms should be defined the first time they appear in each of three sections: the abstract; the main text; the first figure or table. When defined for the first time, the acronym/abbreviation/initialism should be added in parentheses after the written-out form.

Reviewer 3 Report

Comments and Suggestions for Authors

Dynamic Changes in Distribution of Hydrocodone and Oxycodone in Florida from 2006 to 2014

Florida, known for its high number of opioid-prescribing physicians and unregulated pain clinics, experienced unique challenges during the opioid epidemic. In this manuscript, the authors, using data from the Washington Post and the DEA's ARCOS databases, examined the distribution rates of oxycodone and hydrocodone across Florida counties from 2006 to 2014. The data, converted to morphine milligram equivalents (MME), showed a sharp increase in oxycodone distribution from 2006 to 2010, with a significant decline afterward. In 2010, the average MME per person in Florida was 729.4, marking a 120.6% rise from 2006. The counties with the highest MME per person in 2010 were Hillsborough, Hernando, and Broward, significantly exceeding the state average. Legislative measures between 2009 and 2011 correlate with the subsequent reduction in opioid distribution post-2010, highlighting marked differences in opioid distribution across Florida counties during the epidemic's peak.

Comments:

The manuscript is interesting and presents new and useful contexts. The introduction provides sufficient background and the figures provide detailed information. The conclusions are consistent with the evidence and the limitations of this study are already considered by the authors. The references are appropriate and exhaustive. English language and style are fine. 

However, the authors state that there may have been an unintended consequence of the legislation—fentanyl has become a major problem in Florida.

It is requested, therefore, that the authors explain how legislation may have facilitated the spread of fentanyl.

The paper can be accepted after minor revision.

Reviewer 4 Report

Comments and Suggestions for Authors

Please clarify:

  • What are the specific differences between the standard ARCOS data and those obtained from the Washington Post?
  • Why was the Washington Post data chosen in addition to the standard ARCOS data?
  • How was the conversion from grams of oxycodone and hydrocodone to morphine milligram equivalents (MME) performed?
  • What are the limitations and potential biases of using ARCOS data compared to other data sources?
  • Were there any ethical issues identified during the approval of the research protocol?

Please explain:

  • Can the small rebounds in opioid distribution observed between 2016 and 2017 be explained?
  • Can the limitations of the work be further elaborated?

Round 2

Reviewer 2 Report

Comments and Suggestions for Authors

The authors have satisfactorily addressed all my concerns. I consider the manuscript acceptable for publication.